# Impact of climate and land use on the temporal variability of sand fly density in Sri Lanka: A 2-year longitudinal study

**Sanath C. Senanayake**[1‡], **Prasad Liyanage**[2‡], **Dulani R. K. Pathirage**[1], **M. F. Raushan Siraj**[1], **B. G. D. Nissanka Kolitha De Silva**[3], **Nadira D. Karunaweera**[1]*

**1** Department of Parasitology, Faculty of Medicine, University of Colombo, Colombo, Sri Lanka, **2** Department of Research and Evaluation, National Institute of Health Sciences Kalutara, Ministry of Health, Sri Lanka, **3** Center for Biotechnology, Department of Zoology, Faculty of Applied Sciences, University of Sri Jayewardenepura, Nugegoda, Sri Lanka

‡ These authors are joint senior authors on this work.
* nadira@parasit.cmb.ac.lk

**Data Availability Statement:** The data that support the findings of this study are publicly available from

## Abstract

### Background

Leishmaniasis has emerged as an escalating public health problem in Sri Lanka, with reported cases increasing nearly three folds over past decade, from 1,367 in 2014 to 3714 cases in 2023. Phlebotominae sand flies are the vectors of leishmaniasis. Their density is known to be influenced by context-specific climatic and land use patterns. Thus, we aimed to investigate how these factors drive sand fly density across Sri Lanka.

### Methodology/Principal findings

We analysed monthly collections of sand flies (n = 38,594) and weather data from ten sentinel sites representing three main geo-climatic zones across Sri Lanka, over 24 months. Site-specific land use data was also recorded. The influence of climate and land use patterns on sand fly density across the sentinel sites were estimated using distributed lag nonlinear models and machine learning. We found that climate played a major role on sand fly density compared to land use structure. Increase in rainfall and relative humidity at real time, and ambient temperature and soil temperature with a 2-month lag were associated with a statistically significant increase in sand fly density. The maximum relative risk (RR) observed was 3.76 (95% CI: 1.58–8.96) for rainfall at 120 mm/month, 2.14 (95% CI: 1.04–4.38) for relative humidity at 82% (both at real time). The maximum RR was 2.81 (95% CI: 1.09–7.35) for ambient temperature at 34.5°C, and 11.6 (95% CI, 4.38–30.76) for soil temperature (both at a 2-month lag). The real-time increase in ambient temperature, sunshine hours, and evaporation rate, however, reduced sand fly density homogeneously in all study settings. The high density of chena and coconut plantations, together with low density of dense forests, homesteads, and low human footprint values, positively influenced sand fly density.

GitHub repository with the identifier(s) https://github.com/PrasadLiyanage/Sand_fly_SriLanka.git.

**Funding:** This work was supported by the National Institute of Allergy and Infectious Diseases of the National Institutes of Health, USA, under award number U01AI136033 to NDK. The funder had no role in study design, data collection and analysis, decision to publish, or preparation of the manuscript. The content is solely the responsibility of the authors and does not necessarily represent the official views of the National Institutes of Health.

**Competing interests:** The authors have declared that no competing interests exist.

## Conclusions/Significance

The findings improve our understanding of the dynamic influence of environment on sand fly densities and spread of leishmaniasis. This knowledge lays a foundation for forecasting of sand fly densities and designing targeted interventions for mitigating the growing burden of leishmaniasis among the most vulnerable populations, particularly in an era of changing climate.

### Author summary

Leishmaniasis, a public health problem in the tropics is caused by *Leishmania* species parasites and transmitted by sand flies. Both climatic and non-climatic factors may affect sand fly ecology. The goal herein was to understand how these factors influence sand fly density in 10 field sites across Sri Lanka with varying eco-climatic conditions.

Monthly collections of sand flies over 24 months were analysed, and the influence of climate and land use patterns on the sand fly density was estimated. The results indicated that climate has a more significant impact on sand fly density compared to land use factors. An increase in rainfall and relative humidity were associated with a prominent increase in sand fly density. Similar effects were seen with the rise of ambient and soil temperature, and evaporation rates, albeit with a 2-month lag period. The increase in ambient temperature, sunshine hours, and evaporation rate at real-time, however, uniformly reduced sand fly density. Large land areas of chena and coconut plantations, along with sparse forests, homesteads, and reduced human footprint indices, positively influenced sand fly density. The dry zone appeared to have an increased risk of higher sand fly densities compared to the wet zone. The findings promote a better understanding of the changing climatic and environmental influence on sand fly vectors, leishmaniasis spread, and providing a foundation for the development of targeted interventions for vector and disease control.

## Introduction

The subfamily Phlebotominae (sand flies) includes as many as 800 species [1]. Sand flies are small (2 to 3mm in size) heavily setose, hematophagous insects that live in warm tropical and sub-tropical regions between 50˚N and 40˚S [2]. Sand flies can transmit several bacterial, viral, and parasitic diseases, including leishmaniases [3]. Leishmaniases are a group of diseases caused by more than 20 *Leishmania* species of parasites transmitted through the bites of infected female Phlebotominae sand flies [1]. More than 90 sand fly species are known to transmit the parasite with *Phlebotomus* spp. confined to Asia, Africa and the Europe [4]. The type of resultant disease in leishmaniases depends on the causative *Leishmania* species, with clinical manifestations ranging from self-limiting cutaneous lesions to life-threatening visceral disease [1]. The clinical outcome depends on the fine interplay between parasite, vector, and host factors, mainly with the involvement of the immune system [5]. Accordingly, the disease manifestations are in three main forms; visceral (VL), the most serious form; mucocutaneous (MCL), the most disabling and cutaneous (CL), the most common [1]. Between 0.7 to 1 million new cases of cutaneous leishmaniasis occur annually, ranking it third among neglected tropical diseases [6]. Although the disease is endemic in about a hundred tropical and sub-tropical countries, approximately 85% of new cases are concentrated in ten countries: Afghanistan, Algeria, Brazil, Colombia, Iraq, Libya, Pakistan, Peru, the Syrian Arab Republic and

Tunisia [1]. The disease is associated with poverty, poor living conditions, and environmental changes such as deforestation, dam construction, irrigation schemes, and urbanization [6–9].

Leishmaniasis is a climate-sensitive disease since the sand fly vectors are thermophilic, requiring warm temperatures for survival. The developmental stages of these vectors consist of eggs, larvae, pupae, and adults. The immature stages require cool, moist habitat rich in organic matter and therefore, are sensitive to environmental factors, particularly the weather [10]. The immature stages do not require standing water to complete the life cycle. The hatching of eggs is highly dependent on temperature, with first instar larvae emerging 12 to 19 days after oviposition, pupae in 25 to 59 days, and adults in 35 to 69 days [11]. Laboratory studies have shown that extreme temperatures below 15˚C and above 32˚C have a negative impact on the fecundity and longevity of these flies [12]. The influence of weather variables such as rainfall, relative humidity, soil water stress, evaporation rate, wind speed and El Niño Southern Oscillation on the transmission of leishmaniasis had been evaluated in the past across a variety of endemic settings, but the reported associations are diverse and area specific [13–18]. The apparent diversity could be largely due to the type of data and methods used in the analysis, the location-specific influences of the climate on bionomics of the sand fly species and the transmission dynamics of the respective disease entities.

Leishmaniasis has become a significant public health issue in Sri Lanka. In contrast to the declining disease trends observed in other Southeast Asian countries, Sri Lanka has been experiencing a steady increase in case numbers of leishmaniasis with an exponential rise from 35 cases in year 2001 through 1376 cases in 2014 to approximately 3714 cases in 2023 [19,20]. Almost all the leishmaniasis clinical cases in Sri Lanka are CL caused by *Leishmania donovani* [21], except for a few cases of *L.tropica*-induced CL reported more recently [22]. The parasite is probably transmitted through the species *Phlebotomus argentipes glaucus*, which demonstrates zoophilic behavior compared to other related species in India [23,24]. The continuous upsurge of disease transmission in the country warrants urgent attention to design effective control interventions that might enable meeting equivalent elimination targets as established for VL in the region. These targets involve reducing the incidence to less than one case per 10,000 population [25,26]; the targets specified by the WHO roadmap for neglected tropical diseases 2021–2030 [27]. Climate change and related environmental and socio-economic impacts may catalyze the transmission dynamics in future, further aggravating the existing disease burden. Within this context, it is important to understand the intricate relationship between climate, environmental factors, and sand fly densities to face the growing burden of sand fly-borne diseases. The current study describes the distribution of sand flies in different geographic zones in Sri Lanka, related to disease hotspots. It also examines the influence of local weather and non-climate land use patterns on sand fly density, providing insights that are relevant and applicable for planning effective interventions to control leishmaniasis in any endemic country.

## Methods

### Ethics approval and consent to participate

Ethics approval for the study was granted by the Ethics Review Committee, Faculty of Medicine, University of Colombo, Sri Lanka (Ref no. EC-17-062).

### Study areas meteorological and georeferenced land-use data

Sri Lanka is an island with an area of 65,525 km$^2$ located between latitudes 5˚55' and 9˚51'N and longitudes 79˚41 and 81˚53'E. The country is divided into four climatic zones based predominantly on the rainfall, viz. wet zone, intermediate zone, dry zone, and semi-arid zone.

The wet zone, located in the southwest part of the island and central hills, receives the maximum rainfall in the country with an annual average between 2500 to 5000mm. The maximum rainfall occurs during the southwest (SW) monsoon from May to September and the northeast (NE) monsoon from November to January. The dry zone covers most parts of the country and receives an annual rainfall between 1200 and 1900mm during the NE monsoon with little or no rain for the rest of the year. An intermediate zone situated between wet and dry zones in the island receives an average annual rainfall of 1500-2500mm, whereas the semi-arid zones situated within the dry zone of the country receive an average annual rainfall of 800-1200mm [28,29]. The country is divided into 25 districts for administrative purposes, and they are nested within 9 provinces. Nine sentinel sites were strategically chosen to conduct sand fly collections, aiming to closely represent each province and encompass three main climate zones (wet, intermediate and dry). An additional sentinel site, Delft, situated on Delft Island in the Palk Strait, was chosen from the Northern province. The location of the sentinel site within each province was based on the case records of each Medical Officer of Health (MOH) area during the year 2017 as maintained at the Epidemiology Unit, Ministry of Health and also in consultation with the respective Public Health Officials. A perimeter of 5km from the sentinel site was used to study topological factors such as vegetation cover and land use patterns, including water bodies. The human pressure on the study settings was quantified by the Human Footprint Index (HFI) [30,31]. The HFI integrates eight key indicators including built environments, population density, electric infrastructure, crop and pasture lands, roads, railways, and navigable waterways to quantify anthropogenic pressures at a fine spatial resolution (30 arcsec) [30,31]. The ten sentinel sites represented main climate zones of Sri Lanka and were named as per the township in which they were located to, viz. Ambanpola, Dickwella, Delft Island, Welioya, Mamadala, Kataragama, Mahaoya, Mirigama, Peradeniya and Thalawa (Table 1).

## Sand fly collection

The adult sand flies were collected from March 2018 to February 2020 in ten sentinel sites over twenty-four months. Each site was equipped with UV LED CDC light traps (LT) a product of BioQuip, USA and cattle-baited net traps (CBNT) (Fig A in S1 Text). The CBNTs used were 10 x 10 feet in size and a single animal was placed within the trap and kept overnight. Sand fly samples were collected using a manual aspirator at 10 pm and 4 am the following day. The trapping was conducted for two consecutive nights per month using ten LTs and one CBNT per night. The collection time was kept constant as much as possible, for all sites in every visit. Each site was visited approximately on the same day of the month. Day time collections were not done based on previous observations on nocturnal behavior of sandflies [32]. However, in Delft Island, sand fly collections were done only twice a year due to logistical constraints. The CBNT was placed in a constant place while the ten LTs were placed in 20 houses in rotation within a radius of 500m to CBNT. A minimum distance of about 50m was maintained between CBNT and LTs. The distance between the houses ranged from 10m to 200m depending on the area. The Fig B in S1 Text shows the placement of CBNT and LTs in the Kataragama sentinel site. The same methodology was used in all sites including the Delft Island. However, the data from Delft collections were used only for the descriptive analysis and excluded from the time series analysis due to less frequent sampling. Another exception was Peradeniya, where the trapping was done monthly with predominant use of LTs, due to the difficulties in obtaining cattle for CBNTs (only two CBNT cycles were completed). The collected sand flies were preserved in absolute ethanol and transported to the laboratory for further analysis. Species identification of collected sand flies was done based on morphological features using standard keys

**Table 1. Characteristics of the sentinel sites.**

| Province | District | Township and GPS coordinate of site | Altitude category# | Climatic zone## | CL incidence (2018 to 2020)* |
|---|---|---|---|---|---|
| North-western | Kurunegala | Ambanpola 80.2463E/7.89703N | Lowland | Intermediate | 6.97 |
| Southern | Matara | Dickwella 80.7015E/5.97627N | Coastal | Wet zone | 19.26 |
| Northern | Jaffna | Delft island 79.4314E/9.31090N | Coastal | Dry zone | 0.76 |
| Northern | Mullaitivu | Welioya 80.8110E/8.98232N | Coastal | Dry zone | 4.86 |
| Southern | Hambantota | Mamadala 80.9667E/6.17158N | Lowland | Dry Zone | 41.82 |
| Uva | Monaragala | Kataragama 81.3132E/6.42649N | Lowland | Dry zone | 2.86 |
| Eastern | Ampara | Mahaoya 81.3234E/7.48443N | Lowland | Dry zone | 0.45 |
| Western | Gampaha | Mirigama 80.0967E/7.22750N | Lowland | Wet zone | 0.65 |
| Central | Kandy | Peradeniya 80.6007E/7.26643N | Highland | Wet zone | 0.66 |
| North Central | Anuradhapura | Thalawa 80.3400E/8.19928N | Lowland | Dry zone | 25.46 |

#Altitude category: Costal: Surrounds the island with elevation of about 30 m above sea level; Lowland: 30 to 1000m above sea level; Highland: mountainous areas with an elevation of 1000 to 2500 m above sea level.

##Climatic zone: Arbitrary division of the island based on annual rainfall

*Leishmaniasis incidence per 100,000 population

[33]. Forty-eight cattle-baited trap nights and 960 light trap nights were done across the country to collect *P. argentipes* during the study period. The density of *Phlebotomus argentipes* was assessed by calculating the sand fly counts per trap per night for both light traps (LT per trap) and cattle-baited net traps (CBNT per trap). Additionally, the total sand fly counts were calculated for all ten light traps (LT total) and one CBNT (CBNT total), collected over two nights at each sentinel site. Fig C and in S1 Text shows the spatial variability and seasonality and, Table A in S1 Text shows the summary statistics of the sand fly counts averaged across each sentinel site.

## Climate data

Monthly mean rainfall, ambient temperature (minimum and maximum), relative humidity, wind speed, soil temperature (measured at 08:30 and 15:30 hours at 5cm and 10cm depth), evaporation and sunshine hours data from March 2018 to February 2020 were obtained from the Meteorological Department of Sri Lanka. Meteorological stations nearest to the sampling sites were selected using the GPS coordinates of both the surveillance sites and the meteorological monitoring stations. We supplemented the ground-level weather data with ERA5-Land reanalysis climate dataset, which is accessible through the Copernicus Climate Change Service Climate Data Store. [34]. ERA5-Land climate data for rainfall, temperature, wind speed and soil temperature within a 5km buffer around the geolocations of the surveillance sites were downloaded in netCDF file format and processed using R package ncdf4 [35]. The ERA datasets (ERA5 and ERA-Interim) do not directly archive Relative Humidity (RH). Therefore, RH was derived from near-surface temperature and dew point temperature based on the Bolton

formula [36]. Furthermore, information on sunshine hours was only accessible for five study locations (Mamadala, Thalawa, Ambanpola, Kataragama, and Dickwella). The Fig D in S1 Text show the seasonality of climate variables averaged across all study settings and, Fig E in S1 Text shows the correlation between climate variables.

## Non-climate land use information in study settings

Location-specific characteristics, that can modify the relationship between weather variability and sand fly density, were obtained for the 5km buffer area around the surveillance sites. The 5 km radius was chosen based on ground level and remote observations (using ArcGIS) made in consultation with geographic experts to balance capturing relevant environmental variability while avoiding excessive spatial dilution effects. This scale is supported by previous studies that have shown that sand fly populations can be influenced by landscape features and environmental conditions extending several kilometres [37]. Geo-referenced land-use data were further validated by the Sri Lanka Survey Department [38]. The land-use data was clipped and extracted from the 5km buffer around the sentinel site using ArcGIS software. The area of each land-use type was derived using a geometry calculator. The land use values were exported as a database file, which was opened through the Excel application, and the spread of equal land-use categories were totalled using the PivotTable in the Excel application. Land use variables included land areas of paddy fields, dense forests, coconut cultivars, chena cultivars, marshy lands, scrubs lands, rocks, water reservoirs (tanks), streams, water bodies, cemeteries and homesteads. In addition to these variables, we used HFI as a measure of human pressure on land. This dimensionless index calculated as a continuous scale of increasing human pressure from 0 to 50 where more than 12 is considered to be areas with intense human pressure [39]. Furthermore, the HFI provides spatially explicit and temporarily inter-comparable measures of human interaction with the environment and local natural systems. We used the most recent HFI maps, generated up to 2019, which were created using a machine-learning approach based on the original HFI dataset available from 2000 to 2013 [31]. We extracted the average HFI for each study year for a buffer of 5km at each surveillance site. The distribution of these variables among each surveillance site is given in the Table B in S1 Text.

## Leishmaniasis incidence

Leishmaniasis is a notifiable diseases in Sri Lanka with a mandatory regulation in place for case notification to the national integrated communicable disease surveillance system in the country. The number of leishmaniasis cases from March 2018 to February 2020 and the annual average incidence rates of leishmaniasis per 100,000 population by each district were obtained from the Epidemiology Unit, Ministry of Health of Sri Lanka [20].

## Statistical analysis

Here we used a combination of two analytical approaches. Firstly, we utilized distributed lag non-linear models (DLNMs) [40] in a two-staged hierarchical meta-analytical framework [41] to assess the delayed (lagged) association between weather variables and sand fly density across all sentinel sites in Sri Lanka. Secondly, we employed XGBoost, an ensemble boosted decision tree method, to ascertain how these lagged weather variables, along with context-specific land use variables, contribute to sand fly densities across study settings [42]. One of the notable advantages of the XGBoost machine learning algorithms is its capability to handle highly non-linear, correlated, and interactive covariates which cannot be implemented in DLNM framework alone. A summary of the statistical analytical approach used here is provided in Box A in S1 Text.

## Evaluating the lagged influence of climate variables on sand fly densities

The DLNMs implemented in the R package *dlnm* (version number 2.4.6) use the concept of creating flexible cross-basis function estimators to capture simultaneously the delayed and non-linear dependencies of the exposure and outcome data [43]. In the first stage, the exposure-lag-response associations for each study setting were flexibly estimated using ground level and remotely sensed weather data. A quasi-Poisson time series regression model was used to account for the over-dispersion of data and the influence of time-varying confounders. The common formula for the first stage sentinel site-specific models for weather variables and sand fly density measurements is given as

$$VI\ i \sim quasiPoisson(\mu t)$$

$$E(VI_{ti}) = \beta i + f(Weathre_{ti}, vardf, lagdf) + s(T_{ti}, timedf)$$

Where $E(VI_{ti})$ was the expected value for each sand fly density measurements (LT per trap, LT total or CBNT per trap) obtained by LTs and CBNTs in each month $(_t)$ in each surveillance setting $(_i)$. $\beta$ was the intercept, and $f(Weathre_{ti}, vardf, lagdf)$ was the cross-basis function for each weather variable (rainfall, maximum, minimum and mean temperature, soil temperature, relative humidity, sunshine hours etc. respectively, in each model). The *vardf* and *lagdf* were the corresponding degrees of freedom set for weather variables and for their lag values. $s(T_{ti}, timedf)$ was the smooth function of time with the degrees of freedom used to account for the time-varying confounders on the outcome. A lag of up to three months was considered to account for the lifecycle of sand fly vectors, ensuring that all biologically plausible associations between climate variables and sand fly density were captured.

In the second stage, the surveillance site-specific exposure-response associations were meta-analysed to obtain joint estimates for the country accounting for within and in-between surveillance site-level variability. The model output was given as a relative risk (RR) estimate calculated for the full range of exposure values with reference to a risk at a predetermined central reference. We used a multivariate extension to the Cochrane Q-test of heterogeneity to assess the statistical significance of the heterogeneity of the estimates across study setting and it was further quantified by using $I^2$ statistics [44].

The models were evaluated using the quasi-Akaike information criterion (q-AIC) [45]. q-AIC values derived during the model building and selection procedure for each sand fly density measurement is given in the Table C in S1 Text. The lowest q-AIC values observed for LT per trap (sand fly counts per LT per night) indicate the better model fit compared to CBNT per trap (sand fly counts per CBNT per night) and LT monthly total for all weather parameters. Therefore, we used sand fly density values obtained using the LT per trap for our primary analysis and compared the results with that of CBNT per trap where relevant. Definition of the cross-basis functions with respect to different knot positioning for the best-fit models are reported in the Table D in S1 Text. The *mvmeta* package (version number 1.0.3) provided the second stage multi-variate meta-analysis [43]. The divisional heterogeneity of each climate variable is presented in the Table E in S1 Text. To investigate whether the divisional heterogeneity could be explained by different climate zones (wet, dry, and intermediate) and to assess their moderator effect on the observed associations, we extended our analysis by incorporating these climate zones into a univariable multivariate meta-regression framework using the *mvmeta* package [43]. The Wald test was employed to determine the statistical significance of the moderator effect. Further details of this analysis are provided in section 5 of the S1 Text (pages 14 to 15).

### Evaluating the relative contribution of weather and land use variables on sand fly densities

XGBoost, uses gradient boosting which has a comparative advantage over other tree-boosting methods in terms of its versatility, scalability, speed, and optimization to solve complex problems. Recent advancements in machine-learning have led to the development of explanatory frameworks for interpreting the model outputs. These are often referred to as explainable AI (XAI) [46]. We coupled the XGBoost output with the XAI post-processed model interpretation framework, Shapley Additive Explanation (SHAP), which allowed ranking the features of the model (climate and land use variables in the present setting) in their order of contribution [47]. SHAP determines the importance of the feature by comparing a model's predictions with and without a specific feature, considering all possible feature combinations for each observation. The ranking of features is based on their individual contributions for each observation and then averaged across all observations. Details of the XGBoost model building using all weather variable identified by the DLNM approach along with the land use variables is given in the section 6 in S1 Text (Tables G and H in S1 Text). All analytical steps were implemented within the R statistical environment (version 4.1.0) [48].

## Results

### Sand fly species composition and sex ratio

*Phlebotomus argentipes* was the predominant sand fly species captured, accounting for 38,594 sand flies (female: male ratio = 4,246:34,348), (Table 2). The remaining sand flies (n = 333; <1%) belonged to the genus *Sergentomyia*. The female-to-male ratio in the total *P. argentipes* sand fly collection (LT and CBNT together) was approximately 1:8.09 indicating the presence of approximately eight times more males than females in the collection. The female-to-male ratio of *P. argentipes* however, varied depending on the trap type, with a ratio of 1:9.99 in the cattle-baited traps and 1:1.8 in the light traps.

**Table 2. Total counts of *P. argentipes* with sex differentiation and female-to-male ratio recorded using different trap collection methods, and the annual average density of *P. argentipes* in sentinel sites from March 2018 to February 2020.**

| Site | CBNT | | LT | | Annual average density of *P.argentipes* |
|---|---|---|---|---|---|
| | Total | Female to male ratio | Total | Female to male ratio | |
| Ambanpola | 4,619 | 1:5.3 | 358 | 1:1.1 | 2,488.5 |
| Dickwella | 7,720 | 1:17.8 | 353 | 1:5.1 | 4,036.5 |
| Delft Island | 1,917 | 1:21.8 | 29 | 1:0.9 | 5,838** |
| Welioya | 1,691 | 1:5.4 | 467 | 1:1.0 | 1,079 |
| Mamadala | 8704 | 1:9.6 | 387 | 1:1.3 | 4,545.5 |
| Kataragama | 3,825 | 1:15.4 | 305 | 1:2.1 | 2,065 |
| Mahaoya | 1,859 | 1:3.7 | 226 | 1:1.5 | 1,042.5 |
| Mirigama | 3016 | 1:12.5 | 171 | 1:4.0 | 1,593.5 |
| Peradeniya | 72 | 1:5.0 | 166 | 1:3.7 | 515* |
| Talawa | 2,473 | 1:19.8 | 236 | 1:2.9 | 1,354.5 |
| **Total** | **35,896** | **1:9.9** | **2,698** | **1:1.8** | 19,297 |

CBNT: *P. argentipes* collected using cattle-baited net trap

LT: *P. argentipes* collected using CDC light traps

* Based on 2 Cattle-baited net trap (CBNT) and 24 CDC light trap (LT) cycles collections

** Based on 04 CBNT and 04 LT cycles collections

All the other sites the collection is based on 24 CBNT and 24 LT cycles

## Spatial and temporal patterns

The spatial densities of *P. argentipes* captured were highly heterogeneous and variable. Based on the density the sites were arbitrarily classified into High >2500, Mid 1500–2500 and Low <1500 zones. Mamadala, Delft Island and Dickwella were within the high sand fly density zone, whereas Kataragama, Mirigama and Ambanpola were in mid sand fly density zone, and Thalawa, Welioya, Peradeniya and Mahaoya were within the low sand fly zone (Table 2). The sand fly density positively correlated with the leishmaniasis incidence with a tendency for high disease burden areas to record high sand fly densities (Fig 1). However, this pattern was not consistent in all districts with the Spearman's rank correlation coefficient being 0.57,(p-value > 0.088). The temporal variability of the sand fly density showed a clear seasonal pattern peaking from August to September (Fig C in S1 Text).

## Exposure-lag-response associations between weather variables and leishmaniasis vector density

The overall pooled results of the two staged hierarchical meta-analysis using DLNM approach suggested that rainfall, ambient temperature, soil temperature, sunshine hours, mean relative humidity, evaporation and wind speed were associated with leishmaniasis vector density as measured by LT. The different lag dimensions of these variables have varying impact on sand fly density. The exposure-response curves of these weather variables with the corresponding statistically significant lags are given in Figs 2 and 3. The full spectrum of the associations (lag 0 to lag 3) of each weather variable are given in the Figs F-L in S1 Text.

The relative risk of sand fly density associated with CBNT exhibited a similar trend; however, none of the weather variables showed statistically significant associations. The corresponding exposure-lag response curves are presented in Fig M in S1 Text.

## Relative contribution of weather and land use variables on sand fly densities

Fig 4 ranks the twenty predictor variables (weather and land use) based on their SHAP values in descending order. These values elucidate the significance of each variable in influencing sand fly densities, as measured by the light trap (LT per trap) across all surveillance sites. The global feature importance plot illustrates the percentage contribution of each feature, while the local explanation summary demonstrates how these features impact sand fly density across the entire spectrum of values.

Weather variables appeared to be relatively more important for the sand fly densities when compared to the land use variables. Rainfall showed the highest contribution, followed by maximum temperature lag 2, sunshine hours lag 0, maximum temperature lag 0, soil temperature lag 2, wind speed lag 0, relative humidity lag 0 and evaporation lag 0. Out of the land use factors, chena cultivation and dense forest were relatively important compared to other land use variables.

## Rainfall

When pooled across all the sentinel sites, the rainfall appeared to be associated with the risk of increasing sand fly density measured by UV LED CDC trap at lag of 0 (Fig 2A). As shown in the Fig F in S1 Text, when the lag period increases, the exposure-response associations become less obvious. With reference to the risk at a rainfall value of 0, the increase in rainfall was associated with the statistically significant increase in the RR of sand fly density throughout its range of values. The maximum RR of 3.76 with a 95% CI of 1.58 to 8.96 was observed at the

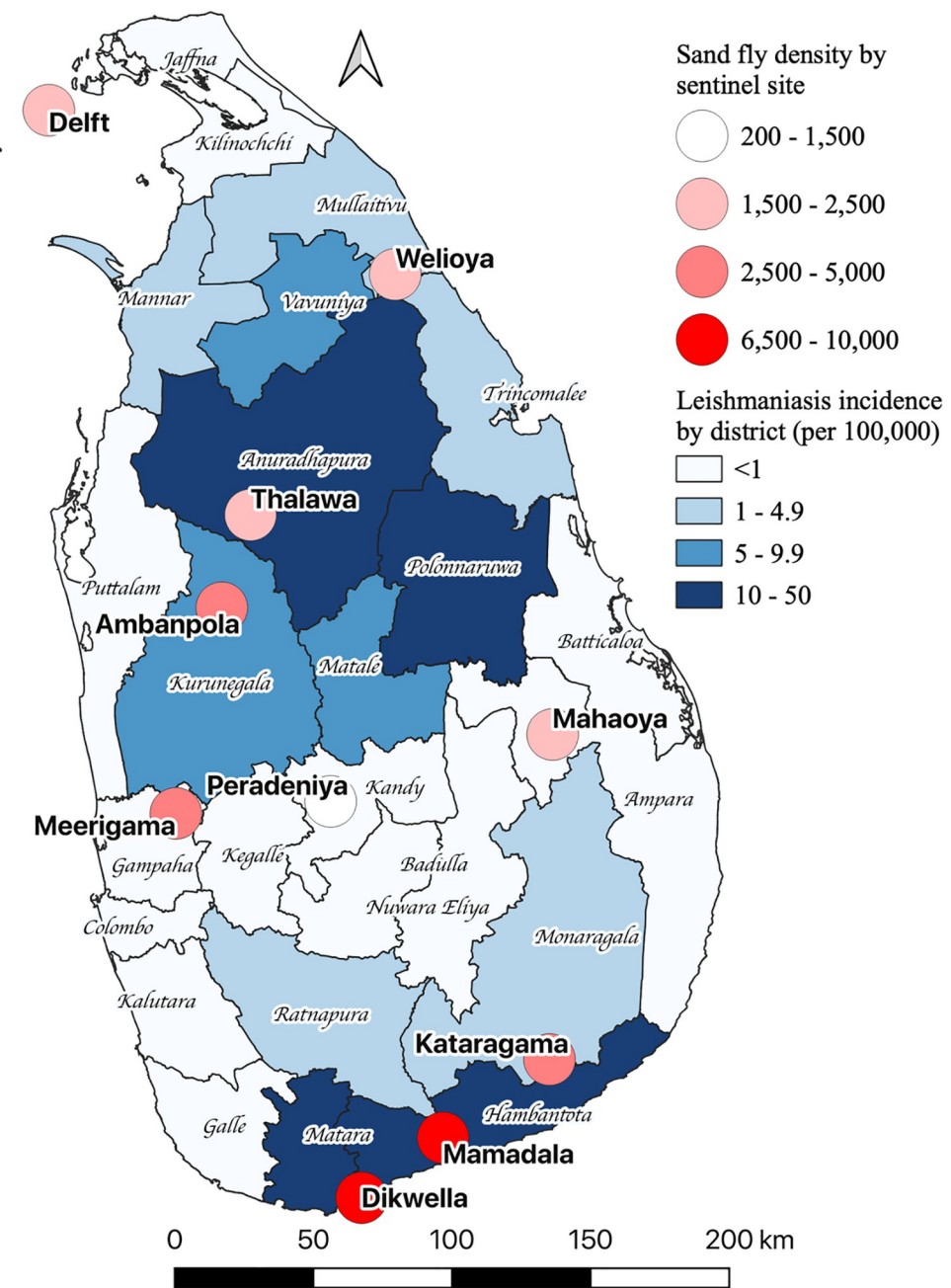

**Fig 1. Sand fly density by sentinel site and annual average leishmaniasis incidence (per 100,000 population) by district from 2018 to 2020 in Sri Lanka.** The blue shaded areas indicate the leishmaniasis incidence and the red shaded circles show the geographical locations and the cumulative number of sand flies collected at each sentinel site for the study period. The base map shapefile was obtained from the Humanitarian Data Exchange. The data is licensed under Creative Commons Attribution International Governmental Organization (CC BY-IGO). No changes were made to the original file. Link to the base layer of the map https://data.humdata.org/dataset/sri-lanka-administrative-levels-0-4-boundaries.

rainfall of 120 mm per month. Thereafter, the RR was observed to slightly decrease with increasing rainfall up to the extreme rainfall value of 524mm per month (RR of 2.83 with 95% CI of 1.12 to 7.14). However, a statistically significant Q test of heterogeneity (p-value 0.003) revealed a substantial variation in the exposure-response association among surveillance sites,

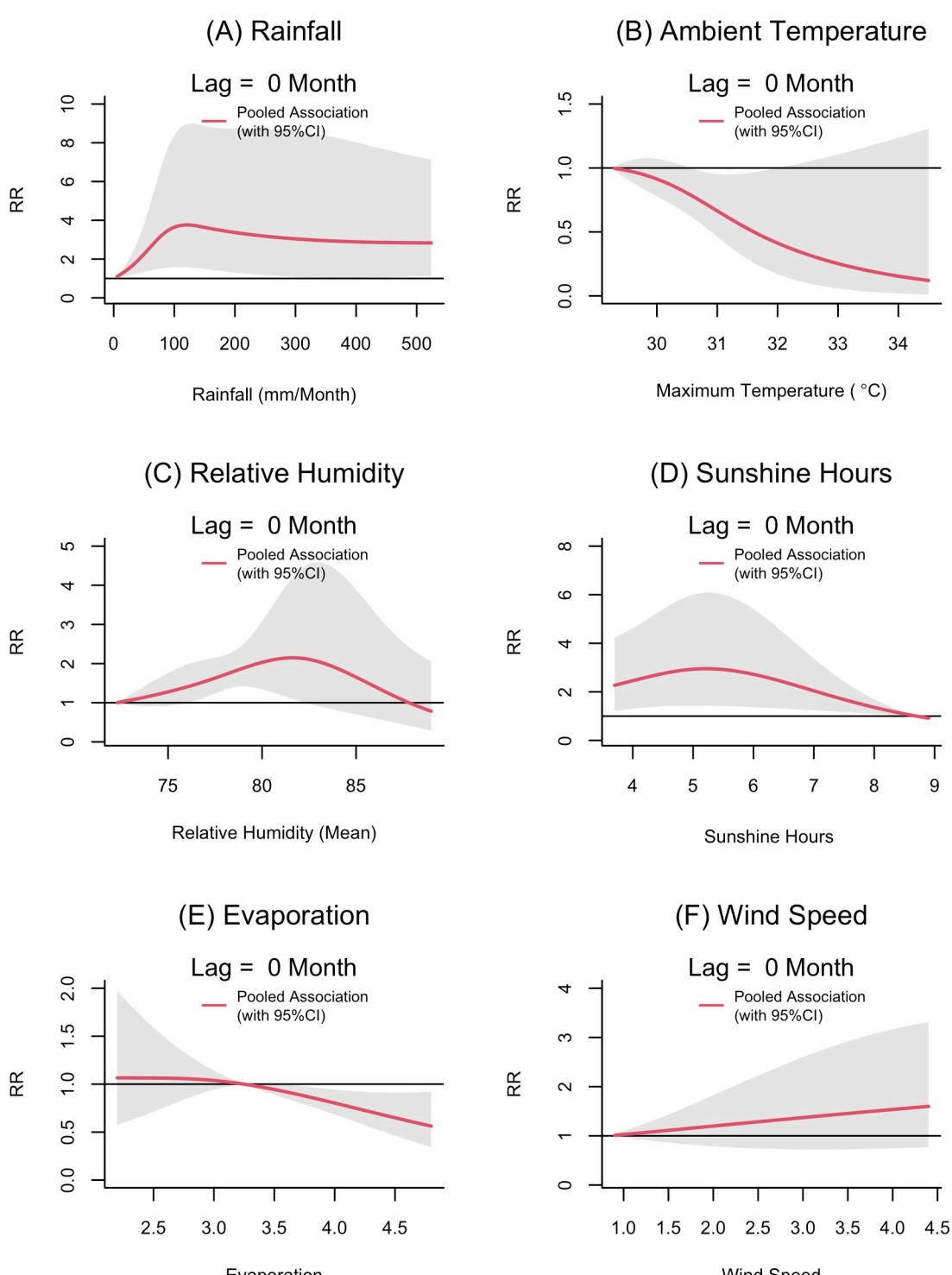

**Fig 2. Relative risk (RR) of leishmaniasis vector density (measured by UV LED CDC traps) by rainfall (A), ambient temperature (maximum temperature) (B), average relative humidity (C), sunshine hours (D), evaporation (E) and wind speed (F) at a lag of 0 months.** The exposure-response functions at lag of 0 month were predicted from the pooled exposure-response function obtained from the meta-analysis for all surveillance sites in Sri Lanka, 2018–20. Shaded areas are 95% CIs. Relative risks were calculated with reference to the risk at a rainfall value of 0 mm per month, maximum temperature of 29.3°C, average relative humidity of 72.25, average evaporation of 3.3mm and wind speed of kmh⁻¹. The most important lags for each exposure variable were selected for presentation. The full spectrum of exposure-lag response associations are given in Fig F-L in S1 Text.

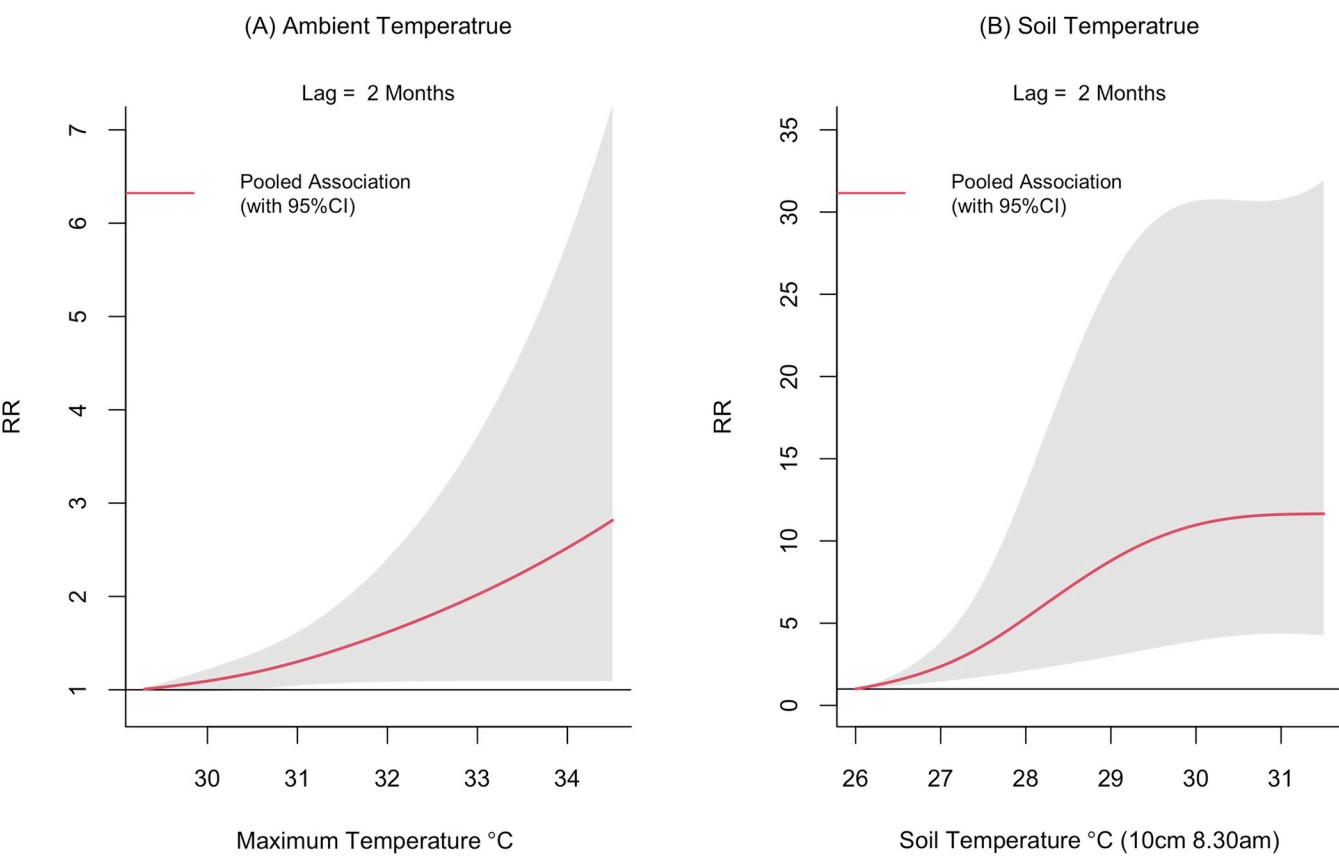

**Fig 3. Relative risk (RR) of leishmaniasis vector density (measured by UV LED CDC traps) by ambient temperature (maximum) (panel A) and soil temperature measured at 8.30 am at 10cm below the surface and evaporation (panel B) at the lag of 2 months.** The exposure-response functions at each lag were predicted from the pooled exposure-response function obtained from the meta-analysis for all surveillance sites in Sri Lanka, 2018–20. Shaded areas are 95% CIs. Relative risks were calculated with reference to the risk at a soil temperature of 26°C and maximum temperature of 29.3°C. The full spectrum of the associations is given in Figs G and K in S1 Text.

with an I statistic of 49%. The local explanation summary (Fig 4B and Fig O in S1 Text) demonstrate that increasing rainfall values predominantly positively influence sand fly densities.

### Ambient temperature

At the lag of 0 months, the increase in ambient temperature (maximum temperature) appeared to reduce the RR of sand fly density (Fig 2B). With reference to the lowest temperature value in the range (29.3°C), the sand fly density appeared to be reduced by each unit increase in the temperature. The associations were statistically significant between 30.6°C to 32°C and the minimum relative risk observed at the temperature value of 34.5°C was 0.12 (95%CI; 0.01 to 1.3). Conversely, the increasing maximum temperature at a lag of 2 months increased the RR of sand fly density and was more influential compared to the lag 0 effect (Figs 3A and 4). The highest RR observed was 2.81(95% CI; 1.09 to 7.35) at 34.5°C. The impact of temperature on sand fly density was homogeneous across all surveillance sites at temperature lag 0, but heterogeneous at lag of 2 (Table E in S1 Text).

### Relative humidity

With reference to the minimum RH value of 72.25, the relative risk of vector density appeared to increase with the increase in RH up to 82 (2.14; 95% CI = 1.04 to 4.38) at the lag of 0 months

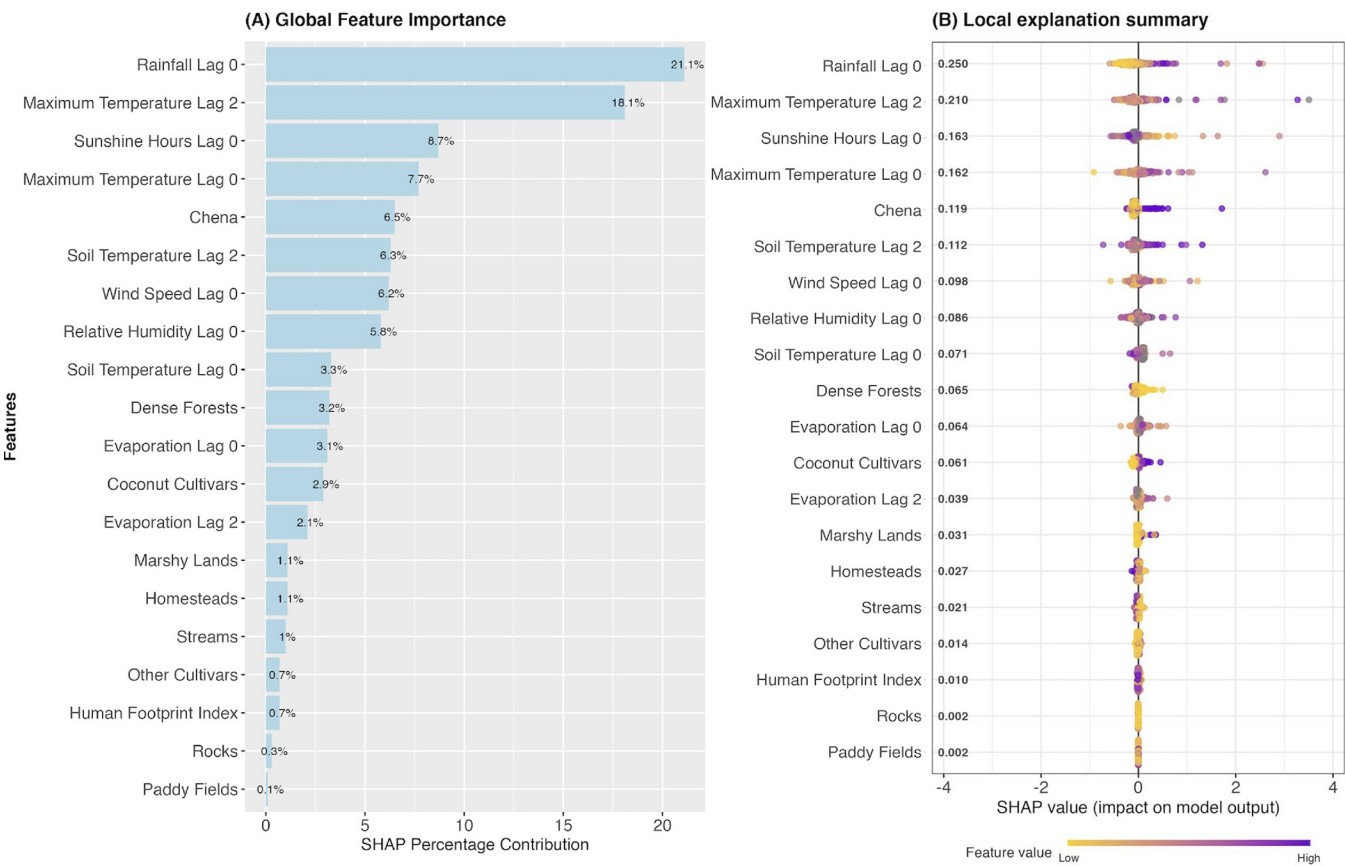

**Fig 4. SHAP feature importance plot for sand fly densities measured by light traps.** The panel A (Global Feature of Importance) bar chart presents the percentage contribution of mean SHAP values estimated for each feature. The panel B (Local Explanation Summary) is a set of beeswarm plots showing feature impact on the model output in their full range of values. The purple color indicates a higher value of corresponding variables and the yellow color indicates a lower value. The dot's position on the x-axis shows the impact that feature on the model's prediction of sand fly densities. Plots of SHAP values for each variable with better visual effect is given in the Figs O and P in S1 Text.

Fig 2C. However, the Q test was statistically significant (p-value 0.012) with $I^2$ of 49.6% indicating substantial heterogeneity among surveillance sites. The SHAP value plot (Fig O in S1 Text) illustrates that increasing RH elevates the RR, with extreme values tending to decrease it.

## Sunshine hours

Increasing sunshine hours appeared to reduce the RR of sand fly densities at a lag of 0 months. The maximum relative risk observed (2.93; 95% CI = 1.43 to 6.0) was at 5 hours of sunshine per day (Fig 2D). When the daily average sunshine hours further increased, the relative risk of vector density appeared to decrease. A similar pattern was observed at a lag of 1 month. The association was not statistically significant, with a further increase in lags (Fig I in S1 Text) The observation was homogeneous across all the settings as suggested by the non-significant Q test (2.77, p-value = 0.950). Our analysis was limited to five surveillance sites (Mamadala, Thalawa, Ambanpola, Katharagama and Dikwella) due to the limited availability of ground and remote sensing data on sunshine hours.

## Wind speed

An increase in wind speed appeared to increase the risk of vector activity at the lag of 0 months (Fig 2F). However, the associations were not statistically significant up to a lag of 3 months

(Fig J in S1 Text). The associations appeared to be homogeneous across sites (Table E in S1 Text). The SHAP value plot (Fig O in S1 Text) illustrates that the extreme values of wind speed have a negative influence on sand fly densities.

### Soil temperature

Increasing soil temperature with a lag of 2 months and measured at 8.30 am at 10cm below the surface was associated with increasing relative risk of sand fly densities (Fig 3B). The risk of vector activity started to increase at a lag of 1 month and reached its maximum at the 2-month lag period before reducing at the lag of 3 months (Fig K in S1 Text)

At a lag of 2 months, the relative risk of sand fly density peaked at 11.6 (95% CI: 4.38 to 30.76) when the soil temperature reached its maximum value of 31.5°C. Similar to the ambient temperature the relative risk of sand fly activity decreased with increasing soil temperature at a lag of 0 months. With reference to the risk estimated at 26°C, the lowest relative risk was observed to be 0.12 (95%CI; 0.03 to 0.40) at a soil temperature value of 31°C. At a lag of 0 months, the observed reduction in the risk of vector activity was homogeneous across study settings as suggested by the non-significant Q test. The heterogeneity was statistically significant at a lag of 2 months (Table E in S1 Text).

### Evaporation

With reference to the evaporation value of 3.25 (which was the median evaporation value observed when averaged across all the settings) the risk of sand fly density appeared to decrease with increasing evaporation at a lag of 0 months when the evaporation value exceeded 3.6 (Fig L in S1 Text). The minimum relative risk observed (0.56; 95% CI = 0.92 to 0.34) was at an evaporation value of 4.8. An opposite pattern was observed at a lag of 2 months where the RR appeared to increase with increasing evaporation. The association was not statistically significant with a further increase in lag periods. The observation was homogeneous across all the settings as suggested by the Q test (Table E in S1 Text, p-value = 0.339). Evaporation appeared to be the least influential weather variable based on the SHAP ranking (Fig 4).

### Influence of the climate zones on the association between weather variables and sand fly density

The analysis indicated that transitioning from wet to dry climate zones was associated with an increased relative risk of sand fly density per unit increase in weather variables. Specifically, in dry zones, the relative risk predicted by rainfall, soil temperature, ambient temperature, and relative humidity was higher than in wet zones (Fig N in S1 Text). Although adjusting for climate zones led to a reduction in the Q-test of heterogeneity, this test remained statistically significant. Additionally, the Wald test did not show statistically significant moderator effects of climate zones on the relationship between weather variables and sand fly density (Table F in S1 Text). This implies that while climate zone differences may partially explain some variability, they do not entirely account for the observed heterogeneity.

### Land use variables

The high land area of chena cultivation, low land areas of dense forests and high land areas of coconut cultivation emerged as important non-climatic factors positively influencing sand fly densities across the surveillance sites. Moreover, other cultivars, marshy lands and paddy fields in comparatively large land areas exhibited a positive influence on sand fly densities. Conversely, a high density of homesteads and high values of the human footprint index and areas

with extensive network of streams were associated with a negative influence on sand fly densities (Figs 4 and P in S1 Text).

## Discussion

The present study aimed to provide a comprehensive analysis of sand fly species distribution and the impact of both weather and land use factors on the sand fly density across diverse geographic and climatic zones in Sri Lanka. By focusing on leishmaniasis transmission hotspots, we sought to quantify how these variables influence sand fly populations in selected sentinel sites. These sites were strategically chosen to represent the range of environmental conditions present in different regions of the country. The temporal variability of sand fly densities was investigated over a period of 24 months through a uniform trap placement across the surveillance sites. Concurrent weather variables viz. monthly average rainfall, ambient temperature, relative humidity, wind speed, soil temperature, evaporation and sunshine hours and land use information collected in proximity to the surveillance site were used to quantify their location-specific influence on the sand fly densities. By combining statistical modelling with machine learning and explainable AI approaches, we identified the climate and land use drivers of sand fly vector density and assessed their relative importance across the country. The results offer insights into the complex interactions between environmental factors and sand fly density, providing valuable information for understanding vector ecology and improving sand fly control strategies in Sri Lanka.

We demonstrate, for the first time, the widespread presence of *P. argentipes* across the country, including in Delft Island. Additionally, our findings support previous reports indicating *P. argentipes* as the predominant species of sand flies in Sri Lanka [49–51]. The sand fly density appeared to differ based on the climatic zone in which the sentinel sites were located. The sex ratio of sand flies collected in this study was significantly biased towards males in the genus *Phlebotomus*, with the effect amplified in CBNT collections. This contrasts with findings from studies conducted under colony conditions, where the sex ratios of emerging flies are approximately equal, despite some species-specific variation [52]. Similar observations of male sand flies outnumbering females in trap collections have been recorded previously both in Sri Lanka and elsewhere, with the behaviour considered analogous to swarming in other Nematocera [53]. The high attractiveness of sand flies to cattle as demonstrated by high counts in CBNTs may be attributed to their preference for animal blood, which is enhanced by its greater body size and $CO_2$/odour output, better suited microenvironment and the availability of the cattle for a sustained and successful blood feed [54,55]. In contrast, CDC LTs are less influenced by such extraneous factors and found to be better correlated with the weather variables.

A positive but not statistically significant correlation was observed between sand fly density and the incidence of leishmaniasis cases recorded from 2018 to 2020. However, this finding may not be surprising since leishmaniasis is a chronic disease, and the manifestation of symptoms typically occurs months or even years after exposure. Moreover, the risk of disease occurrence is influenced by multiple factors beyond vector density, including human immunity, availability of reservoir hosts, human behavior during waking hours, place and hours of sleeping and the type of housing [56–59].

We found that climate is the primary driver of sand fly density across the country, with rainfall, temperature, and sunshine hours playing the most significant roles. Furthermore, weather conditions conducive to sand fly activity are characterised by a combination of moderate rainfall, low sunshine hours, low ambient temperatures, high relative humidity, and low evaporation rates. We noticed a minor decrease in the relative risk (RR) of sand fly activities

during periods of extreme rainfall. Places that are subject to flooding either by soil run off or by direct rainfall are known to be unsuitable for sand fly breeding [4,60]. Once averaged across all study settings, the increasing ambient and soil temperature at real-time (lag zero) negatively correlated with the sand fly activity reducing the relative risk below one. Similarly, laboratory experimental studies have found that increasing temperatures more than 32$^{\circ}$C was associated with higher mortality rates (around 72%) of adult sand flies [12]. However, the ambient temperature (maximum) and soil temperature at a lag of two months exhibited a statistically significant association with an increased risk of sand fly vector density. Remarkably, the ambient temperature with a lag of two months emerged as a highly influential factor, second only to rainfall in its impact on sand fly densities. The studies have found that complete egg to adult development of the sand fly species was temperature-dependent and ranged from 27.89 (+/- 1.88) days at 32$^{\circ}$C to 246.43 (+/- 13.83) days at 18$^{\circ}$C [61]. This time lag between oviposition and emergence of adults correlates with the observed time lag of two months found between the soil temperature and sand fly density in all Sri Lankan study settings, which might be well within the favourable range for egg hatching and larval development. Therefore, it would be reasonable to extrapolate that the exposure to the optimal soil temperatures two months ago may have led to the subsequent emergence of a large number of adults that were attracted to the light traps at the time of surveillance. Increasing mean RH up to 82 during the same month of surveillance may have created a suitable environment for the sand flies to be active. The combined effect of reduced sunshine hours, high evaporation rates, and low humidity at lag zero suggests that sand flies are relatively inactive in sunny and extremely dry conditions. Among the factors investigated, wind speed is likely to have the potential to influence the dynamic behaviour of sand flies, particularly in terms of gene flow between populations without geographical barriers. The gene flow can facilitate the transfer of genes that promote sand fly survival, such as insecticide resistance genes [51] which can have negative implications for vector control programs. We found that extremely high values of wind speed having a negative influence on sand fly density (Fig O in S1 Text).

Among the land use variables measured within a five-kilometre radius from the surveillance sites, cultivation lands have emerged as significant factors influencing sand fly vector densities. Notably, chena cultivars, coconut cultivars, and to some extent, paddy cultivars appeared to play important roles. Alongside other cultivars categorized under broader cultivation lands, these agricultural areas are primarily situated in the dry and intermediate zones of the country. The presence of a low volume of dense forests and streams also suggests conditions typical of the dry zone, potentially contributing to the higher influence on sand fly vector density observed at the lower end of their range. Tree base habitat described for selected species of sand flies might support this observation [60]. The positive moderator effect of dry zone for the weather-sand fly association further suggests the high probability of sand fly presence in the dry zone and subsequently the higher risk of exposure. However, non-agricultural marshy lands, commonly found in the wet and intermediate zones, were also found to have a positive effect. Furthermore, agricultural areas in the dry and intermediate zones in the country typically exhibit lower population densities and reduced human activity compared to urban or residential areas. The current study also indicated that a low number of homesteads and lower values of the Human Footprint Index (HFI) positively influenced sand fly densities. This phenomenon can be attributed to the favourable breeding and resting conditions for sand flies in these less disturbed environments. Agricultural practices may create suitable breeding grounds due to the associated high prevalence of rodents, livestock shelters and irrigation canals [62,63]. The combination of climatic and land use factors appears to creates an environment that supports increasing sand fly activity, a finding that is well-supported by previous studies on breeding habitats of sand flies [64]. Furthermore, a positive and favourable interaction of

the weather variables in the dry zone may be more conducive for the sand fly vectors to thrive and transmit the *Leishmania* parasites. Studies on cutaneous leishmaniasis in the dry zone of Sri Lanka have identified that individuals engaged in occupations such as agriculture, military service, and those with prolonged outdoor exposure (over six hours daily) are at significantly higher risk of infection [65]. This underscores the complexity of interpreting vector density data, which is crucial for evaluating vector control interventions and identifying disease transmission hotspots [66,67].

## Limitations

The density of sand flies in Delft Island was monitored only bi-annually due to logistical constraints and therefore was excluded from the time series analysis. Although the trap placement within a site would have provided the likely influence of the immediate surrounding environment (microenvironment) on sand fly collections, we were unable to record the individual trap data due to logistical constraints. The sand fly collections in each site at each collection time point were pooled, flies counted, identified and presented as per trap per night figures. This limitation however, may have been compensated, at least to some extent by the use of large number of traps per location. The boundary knots of the cross-basis matrices were positioned at the average values of the maximum and minimum values of all division-specific climate variables. This approach aimed to obtain a meaningful estimate for the second-stage meta-analysis. However, it led to limitations in exposure range by excluding extreme values of the respective variables observed in certain surveillance settings. Consequently, the parameter estimates were constrained and did not fully capture the real-world range of exposure. This limitation does not affect the XGBoost approach, which complements the DLNM method by capturing the full range of exposure and addressing the constraints in interpretation. The estimated relative risk values are likely to be context-dependent and may not be directly applicable for all situations. However, the lagged effect is believed to have universal applicability due to its association with the biologically plausible temporal dynamics of sand fly vector life cycles.

## Public health implications of the findings

The findings are significant in predicting vector density in time and space and designing effective strategies to curtail leishmaniasis transmission in a given setting during an era of escalating concern over climate change. This study, while adding to the evidence linking leishmaniasis incidence with changes in environmental factors, provides novel information on the likely effect of selected environmental factors on developing or immature stages of sand flies within the soil, with the resultant lag effect observed on adult sand fly densities. The correlation between sand fly density and key weather variables such as rainfall, temperature, humidity, and sunshine hours, offers a foundation for developing predictive models that can forecast periods of high vector activity. Such models could enable health authorities to implement timely and more effective vector control interventions, potentially reducing disease transmission. Additionally, integrating these environmental insights with epidemiological and demographic data allows for more precise identification of vulnerable populations, improving targeted interventions. Further research endeavours aimed at assessing the impact of environmental factors including wind speed, influence of non-climatic factors on vector density such as human, animal and other variables in the immediate environment and analysis of the soil conditions may provide valuable insights to aid the combat of future public health challenges and for the development of location-specific strategic plans for disease control.

Our analysis identifies Mamadala and Ambanpola in the lowland dry zone, and Dickwella in the coastal wet zone, as high-risk areas due to their significantly elevated sand fly densities.

These regions are characterized by extensive land use for chena, coconut, and paddy cultivation, along with cattle farming and the potential presence of other reservoir hosts. To effectively prevent and control leishmaniasis in similar areas in the country, an integrated One Health approach is essential. This approach should include advanced surveillance tools, such as smart traps for real-time monitoring of vector densities and the surrounding micro environment [68]. Furthermore, analysing pathogens and hosts through sand fly blood meals will provide deeper insights into the host preference and pathogen prevalence which are important for integrated disease control interventions [69]. Active disease surveillance targeting individuals involved in chena cultivation, cattle farming, and other high-risk activities is crucial for effective leishmaniasis control. Public health strategies should integrate interventions that account for agricultural practices, livestock management, and community education on personal protective measures, especially in high-risk areas such as the dry and intermediate zones in the country. Overall, these findings demand a regionally-coordinated strategic plan to address the apparent threat of increasing risk of leishmaniasis, particularly in the face of changing climate. The use of innovative vector surveillance tools, combined with advanced data analytics, can greatly enhance the early detection of leishmaniasis transmission hotspots and support the design of targeted control interventions. Moreover, the potential influence of climate change on vector ecology underscores the urgency for climate-adaptive and integrated disease control strategies. As environmental conditions shift, sand fly distribution and density may change, potentially increasing the risk of leishmaniasis transmission in previously unaffected areas. This calls for the establishment of climate-resilient public health systems that can rapidly adapt to changing vector dynamics. Such an effort may increase the chance of achieving the WHO 2030 targets for effective control and elimination of NTDs in the region.

## Conclusions

This study identifies rainfall, temperature, and sunshine hours as the primary drivers of sand fly density in Sri Lanka. Furthermore, variations in land use patterns and reduced human activity significantly affect sand fly populations. Integrating these findings with epidemiological and demographic data, alongside robust One Health surveillance systems will be crucial for disease outbreak prediction and prevention. This holistic approach, which incorporate a comprehensive understanding of the environmental factors and the ecology of leishmaniasis, will refine existing approaches and develop more accurate disease outbreak predictions capable of targeting the most vulnerable populations. This approach will enable effective leishmaniasis prevention and control in Sri Lanka and beyond.

## Supporting information

**S1 Text. Section 1: Sand fly collection.** Fig A. Sand fly trap types. Fig B. Placement of the light traps and cattle bated traps. **Section 2: Exploratory data analysis**. Fig C. Spatial variability and seasonality of sand fly counts. Table A. Summary statistics of monthly sand fly density by sentinel site. Fig D. Box plots showing the seasonality of climate variables averaged across all surveillance settings from March 2018 to February 2020. Fig E. Correlation between each climatic covariate averaged across all nine monitoring stations from March 2018 to February 2020 in Sri Lanka. Table B. Distribution of non-climate variables among surveillance sites. **Section 3: Statistical analytical approach**. Box A: Statistical Analytical Approach. **Section 4: Distributed Lag Nonlinear Models (DLNM)**. Table C. Sum of the Q-AIC values obtained by the first stage models for LT per trap, CBNT per trap and LT monthly total for each weather variable evaluated. Table D. Definitions of the cross-basis matrix for the selected first stage models. Fig F. Relative risk (RR) of leishmaniasis vector activity (measured by LT) by rainfall at a lag of

0 to 3 months. Fig G. Relative risk (RR) of leishmaniasis vector activity (measured by LT per trap) by ambient temperature (maximum temperature) at a lag of 0 to 3 months. Fig H. Relative risk (RR) of leishmaniasis vector activity (measured by LT per trap) by average relative humidity at a lag of 0 to 3 months. Fig I. Relative risk (RR) of sand fly vector density (measured by LT) by sunshine hours at lag of 0 to 3 months. Fig J. Relative risk (RR) of sand fly vector density (measured by LT per trap) by wind speed at a lag of 0 to 3 months. Fig K. Relative risk (RR) of sand fly vector density (measured by LT per trap) by soil temperature measured at 10 cm from the surface in morning hours (8:30 am) at a lag of 0 to 3 months. Fig L. Relative risk (RR) of sand fly vector density (measured by LT per trap) by evaporation values at a lag of 0 to 3 months. Table E. Quantification of divisional heterogeneity of the association between weather variables and the LT per trap index obtained by the second stage multi-variate meta-analysis. Fig M. Weather and sand fly density measured by CBNT. **Section 5: Moderator effect of climate zones on weather-Sand Fly association**. Table F. Wald test statistics for the moderator effect of climate zones on weather-sand fly association. Fig N. Moderator effect of climate on weather and Sand Fly density. **Section 6: Machine learning and XAI**. Fig O. Plots of SHAP values for climate variables. Fig P. Plots of SHAP values for land use variables. (DOCX)

## Acknowledgments

We acknowledge Mr. Sunil Shantha and Mr. M. P. Ariyapala for field assistance to collect sand flies, and the Head and staff of the Parasitic Diseases Research Unit, Department of Parasitology, Faculty of Medicine, University of Colombo for logistical help.

## Author Contributions

**Conceptualization:** Nadira D. Karunaweera.

**Data curation:** Prasad Liyanage.

**Formal analysis:** Prasad Liyanage, Dulani R. K. Pathirage.

**Funding acquisition:** Nadira D. Karunaweera.

**Investigation:** Sanath C. Senanayake, Dulani R. K. Pathirage, M. F. Raushan Siraj, Nadira D. Karunaweera.

**Methodology:** Sanath C. Senanayake, Prasad Liyanage, M. F. Raushan Siraj, B. G. D. Nissanka Kolitha De Silva, Nadira D. Karunaweera.

**Project administration:** Nadira D. Karunaweera.

**Resources:** Nadira D. Karunaweera.

**Supervision:** Nadira D. Karunaweera.

**Validation:** Nadira D. Karunaweera.

**Writing – original draft:** Prasad Liyanage, Dulani R. K. Pathirage.

**Writing – review & editing:** B. G. D. Nissanka Kolitha De Silva, Nadira D. Karunaweera.

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
