## [Decision Letter · Decision Letter 0]

25 Jun 2024

Dear Prof. Karunaweera,

Thank you very much for submitting your manuscript "Impact of climatic factors on the temporal variability of sand fly abundance in Sri Lanka: A 2-year longitudinal study" for consideration at PLOS Neglected Tropical Diseases. As with all papers reviewed by the journal, your manuscript was reviewed by members of the editorial board and by several independent reviewers. In light of the reviews (below this email), we would like to invite the resubmission of a significantly-revised version that takes into account the reviewers' comments. 

We cannot make any decision about publication until we have seen the revised manuscript and your response to the reviewers' comments. Your revised manuscript is also likely to be sent to reviewers for further evaluation.

Sincerely,

Fabiano Oliveira

Guest Editor

Hira Nakhasi

Section Editor

Reviewer's Responses to Questions

**Key Review Criteria Required for Acceptance?**

**Methods**

-Are the objectives of the study clearly articulated with a clear testable hypothesis stated?

-Is the study design appropriate to address the stated objectives?

-Is the population clearly described and appropriate for the hypothesis being tested?

-Is the sample size sufficient to ensure adequate power to address the hypothesis being tested?

-Were correct statistical analysis used to support conclusions?

-Are there concerns about ethical or regulatory requirements being met?

Reviewer #1: The statistical analysis was impressive and can serve as model for analyses of ecological risk data. The choice of sampling methods and sites were reasonable and well described. The limitations of placement of traps might have been discussed, but the large number of traps per location will have compensated.

Descriptions of the many levels of statistical analysis became difficult to follow, however. A table or even a flow chart can clarify name of program, data analyzed, question answered.

Reviewer #2: Overall, the methods are well-described. I have some questions though, which are listed in the general comments.

Reviewer #3: Yes to all questions

**Results**

-Does the analysis presented match the analysis plan?

-Are the results clearly and completely presented?

-Are the figures (Tables, Images) of sufficient quality for clarity?

Reviewer #1: Excellent maps and clear graphs contribute to visualizing Sri Lanka, trap placement and fly data. Use of commas to indicate 'thousands' is recommended. Plan = analysis.

Graphic indication of effect of each variable on RR was quite interesting.

A table of ranges for each of the 10 sites will help to indicate the degree to which trap placement influences sand fly attraction to the trap (SD perhaps).

Reviewer #2: Overall, the Results are presented in a satisfactory fashion.

Reviewer #3: Analysis followed the analysis plan, results and figures are complete (Table 1 is mroe suitable for the supplement)

**Conclusions**

-Are the conclusions supported by the data presented?

-Are the limitations of analysis clearly described?

-Do the authors discuss how these data can be helpful to advance our understanding of the topic under study?

-Is public health relevance addressed?

Reviewer #1: Conclusions supported by data. Limitations of data were addressed to some extent. Undoubtedly, very large trap-to-trap variation existed within each of the 10 sites, this should have been expressed in a table of ranges to indicate that trap placement and very local environment influences sand fly attraction, and hence inferences of density.

The life cycle of the fly was indicated in the introduction; but the potential effect of each of the climatic variables was not indicated. For example, because eggs and larvae develop in the soil (evidence for depth from literature?), the soil temps are important--as was finally noted in the conclusions. What's the significance of wind -- e.g., adults cannot fly well.

--The unbalanced sex ratio in light traps seems to indicate a serious sampling error. Detailed explanations are expected.

Reviewer #2: The conclusions are supported by the data. The limitations of the study are highlighted.

Reviewer #3: Conclusions are supported by the data, limitations described, authors discuss public health relevance and new insights (these require some sharpening in light of the literature in the field, detailed comments go in the attached commented pdf)

**Editorial and Data Presentation Modifications?**

Reviewer #1: See uploaded listing.

Reviewer #2: Please check the general comments.

Reviewer #3: Please see attached PDF for suggestions about necessary modifications.

**Summary and General Comments**

Reviewer #1: (No Response)

Reviewer #2: In the manuscript, entitled “Impact of climatic factors on the temporal variability of sand fly abundance in Sri Lanka: A 2-year longitudinal study”, by Torres et al., submitted to PNTD in May 2024 (PNTD-D-24-00582), the authors propose to study the dynamic influence of environment on sand flies and leishmaniasis spread in Sri Lanka at the national level. This is an ambitious and interesting study. Overall, it reads well. Below, I list a few reservations in a point-by-point manner. 

Major issues:

1. The authors mention in the limitations, that probably a generalization (which the authors tend to assume) cannot be done. There are a more than a few variables associated with heterogeneity. This is in line with the heterogeneous climate in Sri Lanka: “four climatic zones based predominantly on the rainfall, viz. wet zone, intermediate zone, dry zone, and semi-arid zone”. Shouldn’t you have in account these different zones when you look for associations between “between weather variables and leishmaniasis vector indices”? Can you do the same exercise for each of the different climatic zones?

2. While reading the results, I was under the impression you only used the UV LED CDC trap-derived sandfly density in the association analysis (e.g. “When pooled across all the sentinel sites, the rainfall appeared to be associated with the risk of increasing sand fly density measured by UV LED CDC trap at lag of 0”). Why did you exclude the collection data obtained in the context of CBNTs, via which the majority of sandflies were collected in this study?

3. It would be interesting to understand whether, and to what extent, the different climatic parameters correlate (e.g. rainfall and humidity, temperature and humidity…). Can you do that kind of analysis?

4. The collection methods need further details. Did you collect in arbitrary days regardless of the weather? Did you collect always at the same time? Thinking on “daylight hours”, did you always collect when it was dark?

5. As per the non-climatic variables, I have some doubts on whether you can really measure a real impact, since I am assuming the collection was made under equal conditions (one ecotope selected and used throughout study sites) and you are considering a fairly large 5 Km perimeter. Was this choice based on previous similar studies?

6. The discussion could be complemented with comparisons with other similar studies (in different geographies). Additionally, you could also try discussing why there is a huge difference in the sex ration when you use different collection methods.

Minor points:

1. Line 43: Please end the sentence with a final mark.

2. Line 46-48: There is something wrong in this sentence. You mention 3 RR values but only 2 variables.

3. Line 108: why are you restricting to the genus Phlebotomus?

4. Table 1: You did not study any semi-arid zone? If yes, please mention it in the table.

5. Table 1: Please indicate the actual incidence of CL in each of the provinces studied (instead of only yes/rare).

6. Line 190: Did you always use the same species of animal? Do you know if sandflies naturally chose that species as preferential blood-source?

7. Line 195: Were the light traps placed inside the houses?

8. Line 239: What do you mean with “reservoirs”?

9. Table 2: Please include the sex ratio for each collection site and method (individual by district and by LT and CBNT.

10. Line 372: Do you have enough data to extrapolate that the Delft Island is a “high sandfly density zone”?

Reviewer #3: The manuscript "Impact of climatic factors on the temporal variability of sand fly abundance in Sri Lanka: A 2-year longitudinal study" is a valuable longitudinal ecological study of sand flies which adds information about specific patterns observed in Sri Lanka. The data and analyses seem sound, but some inconsistencies and minor errors needs to be resolved, as well as minor methodological clarifications. In the attached commented pdf comments about errors, missing information and missing additional key references are included. Please use "Sand Fly" consistently and also refer to Sand fly "abundance", not burden.

PLOS authors have the option to publish the peer review history of their article (what does this mean?). If published, this will include your full peer review and any attached files.

Reviewer #1: No

Reviewer #2: No

Reviewer #3: No
---

## [Decision Letter · Decision Letter 1]

4 Nov 2024

Dear Prof. Karunaweera,

We are pleased to inform you that your manuscript 'Impact of climate and land use on the temporal variability of sand fly density in Sri Lanka: A 2-year longitudinal study' has been provisionally accepted for publication in PLOS Neglected Tropical Diseases.

Best regards,

Fabiano Oliveira

Guest Editor

Hira Nakhasi

Section Editor

Shaden Kamhawi

co-Editor-in-Chief

Paul Brindley

co-Editor-in-Chief

Reviewer's Responses to Questions

**Key Review Criteria Required for Acceptance?**

**Methods**

-Are the objectives of the study clearly articulated with a clear testable hypothesis stated?

-Is the study design appropriate to address the stated objectives?

-Is the population clearly described and appropriate for the hypothesis being tested?

-Is the sample size sufficient to ensure adequate power to address the hypothesis being tested?

-Were correct statistical analysis used to support conclusions?

-Are there concerns about ethical or regulatory requirements being met?

Reviewer #2: The Methods are clear. I assume the same is true for the supplementary materials (I could not find the file attached).

Reviewer #3: (No Response)

**Results**

-Does the analysis presented match the analysis plan?

-Are the results clearly and completely presented?

-Are the figures (Tables, Images) of sufficient quality for clarity?

Reviewer #2: The results are clear and explained in a logical matter.

Reviewer #3: (No Response)

**Conclusions**

-Are the conclusions supported by the data presented?

-Are the limitations of analysis clearly described?

-Do the authors discuss how these data can be helpful to advance our understanding of the topic under study?

-Is public health relevance addressed?

Reviewer #2: The conclusions are supported by the data.

Reviewer #3: (No Response)

**Editorial and Data Presentation Modifications?**

Reviewer #2: Accept

Reviewer #3: (No Response)

**Summary and General Comments**

Reviewer #2: The authors addressed my concerns appropriately. The data are impressive, as is the analysis done. Congratulations!

Reviewer #3: Comments were satisfactorily handled.

PLOS authors have the option to publish the peer review history of their article (what does this mean?). If published, this will include your full peer review and any attached files.

Reviewer #2: No

Reviewer #3: No

---

## [Editor Report · Acceptance letter]

14 Nov 2024

Dear Prof. Karunaweera,

We are delighted to inform you that your manuscript, "Impact of climate and land use on the temporal variability of sand fly density in Sri Lanka: A 2-year longitudinal study," has been formally accepted for publication in PLOS Neglected Tropical Diseases.

Best regards,

Shaden Kamhawi

co-Editor-in-Chief

Paul Brindley

co-Editor-in-Chief
